# Altered Extracellular Vesicle miRNA Profile in Prodromal Alzheimer’s Disease

**DOI:** 10.3390/ijms241914749

**Published:** 2023-09-29

**Authors:** Caterina Visconte, Chiara Fenoglio, Maria Serpente, Paola Muti, Andrea Sacconi, Marta Rigoni, Andrea Arighi, Vittoria Borracci, Marina Arcaro, Beatrice Arosio, Evelyn Ferri, Maria Teresa Golia, Elio Scarpini, Daniela Galimberti

**Affiliations:** 1Department of Biomedical, Surgical and Dental Sciences, University of Milan, 20122 Milan, Italy; caterina.visconte@unimi.it (C.V.); paola.muti@unimi.it (P.M.); marta.rigoni@unimi.it (M.R.); mariateresagolia89@gmail.com (M.T.G.); daniela.galimberti@unimi.it (D.G.); 2Neurodegenerative Diseases Unit, Fondazione IRCCS Ca’ Granda, Ospedale Maggiore Policlinico, 20122 Milan, Italy; maria.serpente@policlinico.mi.it (M.S.); andrea.arighi@policlinico.mi.it (A.A.); vittoria.borracci@policlinico.mi.it (V.B.); marina.arcaro@policlinico.mi.it (M.A.); elio.scarpini@unimi.it (E.S.); 3Dental and Maxillo-Facial Surgery Unit, Fondazione IRCCS Ca’ Granda, Ospedale Maggiore Policlinico, 20122 Milan, Italy; 4UOSD Clinical Trial Center, Biostatistics and Bioinformatics, Regina Elena National Cancer Institute—IRCCS, 00144 Rome, Italy; sacconiandrea@hotmail.com; 5Department of Clinical Sciences and Community Health, University of Milan, 20122 Milan, Italy; beatrice.arosio@unimi.it; 6Geriatric Unit, Fondazione IRCCS Ca’ Granda, Ospedale Maggiore Policlinico, 20122 Milan, Italy; evelyn.ferri@policlinico.mi.it; 7National Research Council of Italy, Institute of Neuroscience, Via Raoul Follereau 3, 20854 Vedano al Lambro, Italy

**Keywords:** extracellular vesicles, Alzheimer’s disease (AD), miRNA, biomarker

## Abstract

Extracellular vesicles (EVs) are nanosized vesicles released by almost all body tissues, representing important mediators of cellular communication, and are thus promising candidate biomarkers for neurodegenerative diseases like Alzheimer’s disease (AD). The aim of the present study was to isolate total EVs from plasma and characterize their microRNA (miRNA) contents in AD patients. We isolated total EVs from the plasma of all recruited subjects using ExoQuickULTRA exosome precipitation solution (SBI). Subsequently, circulating total EVs were characterized using Nanosight nanoparticle tracking analysis (NTA), transmission electron microscopy (TEM), and Western blotting. A panel of 754 miRNAs was determined with RT-qPCR using TaqMan OpenArray technology in a QuantStudio 12K System (Thermo Fisher Scientific). The results demonstrated that plasma EVs showed widespread deregulation of specific miRNAs (miR-106a-5p, miR-16-5p, miR-17-5p, miR-195-5p, miR-19b-3p, miR-20a-5p, miR-223-3p, miR-25-3p, miR-296-5p, miR-30b-5p, miR-532-3p, miR-92a-3p, and miR-451a), some of which were already known to be associated with neurological pathologies. A further validation analysis also confirmed a significant upregulation of miR-16-5p, miR-25-3p, miR-92a-3p, and miR-451a in prodromal AD patients, suggesting these dysregulated miRNAs are involved in the early progression of AD.

## 1. Introduction

Prompt clinical diagnosis of Alzheimer’s disease (AD) in its early stage is still uncertain, and unfortunately most patients are diagnosed when they have already progressed to the moderate or severe stages of the disease [1]. At present, the use of the cerebrospinal fluid (CSF) biomarkers amyloid-β (Aβ), tau, and phosphorylated tau (Ptau) allows discrimination between Mild Cognitive Impairment (MCI) due to AD, i.e., prodromal AD, and MCI due to other causes (non-AD MCI) with very high accuracy (see [2] for a review), but this requires an invasive procedure. Therefore, peripheral biomarkers reflecting pathogenic mechanisms occurring in the brain are needed.

In this scenario, extracellular vesicles (EVs), which are important mediators of cellular communication, are promising tools for biomarker discovery. In particular, EVs isolated from the plasma of AD patients have emerged as a suitable potential biomarker of the pathology [3,4,5,6]. EVs are nanosized vesicles of different origins that are released by almost all biological cells in the body [7,8,9]. The nature and abundance of EV contents can vary widely depending on the process of biogenesis, the cell type, and the physiological and pathological states of the donor cell. Additionally, EVs are able to transfer parts of their cargoes to both adjacent and distant cells, whereby they achieve various physiological and pathological functions [9,10]. EV contents include proteins; lipids; and nucleic acid molecules in the form of DNA, mRNA transcripts, and noncoding RNAs (long noncoding RNAs, microRNA, and circular RNA) [7,11].

It is notable that the class of microRNA (miRNA) has recently gained particular attention, both because they are encapsulated in EVs that are remarkably stable in body fluids [11,12] and because a growing body of evidence has shown specific miRNAs or sets of miRNAs associated with certain pathological conditions, including neurodegenerative diseases such as AD [13,14]. Indeed, during the course of some dementias, the expression of many miRNAs was found to be altered [14,15,16,17,18]. Our previous data, considering total plasma EVs, demonstrated increased levels of miR-146b-5p, miR-181a-3p, miR-24-3p, miR-125a-5p, lep-7b-5p, miR-185-3p, and miR-27a-5p in AD patients [19]. In addition, there have been some studies demonstrating a unique miRNA expression pattern in AD patients, while other studies have shown overlapping miRNA signatures. This miRNA expression complexity may be due to epigenetic and environmental factors that are highly variable among different human populations or due to genotypic and phenotypic variation from individual to individual, including age, gender, life style, and the status of AD-related alleles (ApoE and BACE) [15,20,21].

Given the above premises, miRNAs could represent a challenging field for the investigation of innovative and non-invasive biomarkers. In this study, we examined changes in EV-derived miRNAs in the context of dementia through the characterization of the RNA contents in total EVs from the plasma of subjects with or without cognitive decline in order to assess their role as suitable biomarkers for the early diagnosis of AD.

## 2. Results

### 2.1. Characterization of Total EVs from Plasma

The plan of the study comprised two parts: (1) a discovery phase including 11 AD patients, 19 non-AD MCI subjects, and 20 CTRLs tested for 754 candidate miRNAs and (2) a validation phase involving samples from 15 AD patients, 14 prodromal AD patients, 22 non-AD MCI subjects, and 21 CTRLs for the validation of the previously determined best miRNAs. The demographic and clinical information of all participants is summarized in Table 1 and Table 2. First, we characterized the presence of specific surface markers, size, and morphology of total EVs isolated from frozen plasma samples, as suggested by the ISEV [9].

As we and other research groups recently described [22,23], there were no differences in EV concentration or size between AD and CTRL. Approximately 6.46 X1010 ± 4.42 X109 particles/mL could be extracted from 0.25 mL of plasma in an Exoquick ULTRA pellet with a relatively small size, ranging from 90 to 160 nm (Figure 1a). Additionally, the EV samples were subjected to a transmission electron microscope analysis (TEM), confirming the presence of heterogeneous vesicles with the expected size and morphology (Figure 1c). Then, we further characterized, with Western blotting, our total EVs by staining for two EV markers: tetraspanin CD81 and tumor susceptibility gene 101 (TGS101, a component of the ESCRT-1 complex and a vesicular trafficking precursor regulator) (Figure 1b). Moreover, to confirm the specificity of the isolation method used, we used the Golgi marker GM130 as a negative control.

### 2.2. MiRNA Expression Profile in Total Plasma EVs

RNA from total plasma EVs was initially isolated from 11 AD patients, 19 non-AD MCI patients, and 20 CTRL subjects, and the quality and integrity of the extracted RNA were verified using an Agilent 2100 Bioanalyzer (Figure 2a). The amount of RNA detected in our samples was very low, with an approximate average of 554.4 ± 104 pg.

Afterwards, miRNAs were profiled with RT-qPCR using the TaqMan Open Array miRNA panel, which enables the testing of 754 human miRNAs. RNU44, RNU48, and U6snRNA were used as endogenous controls (raw data provided in Appendix A). As shown in Figure 2b, we were able to perform a statistical analysis of about 12% of all detectable miRNAs. Specifically, to ensure sufficient statistical power, we took the step of excluding miRNAs that were undetectable in more than five samples for each subgroup from further analysis. After this filtering process, 162 miRNAs expressed in at least 80% of the samples remained. This approach allowed us to focus our statistical analyses on the most reliable miRNAs. To assess significant differences in the modulation of the expressed miRNAs, we then applied non-parametric statistical tests for multiple group comparisons. Of the 162 miRNAs analyzed, 13 were found to be deregulated in AD patients compared to CTRLs.

Specifically, significantly increased relative expression levels of miR-106a-5p, miR-16-5p, miR-17-5p, miR-195-5p, miR-19b-3p, miR-20a-5p, miR-223-3p, miR-25-3p, miR-296-5p, miR-30b-5p, miR-532-3p, miR-92a-3p, and miR-451a were observed in AD patients compared to CTRLs (details of fold regulation and *p* values are provided in Table 3). Interestingly, some upregulated miRNAs in AD patients belonged to the deeply investigated miRNA 17-92a cluster (miR-17, miR-19b, miR20a, and miR92a) and to its two paralogues: the miR106a-363 (miR-106a, miR-20b, and miR-92a) and miR106b-25 (miR25) clusters [24,25,26]. Therefore, for the validation phase, we only considered some candidate miRNAs for each cluster: miR-106a-5p, miR-16-5p, miR-223-3p, miR-25-3p, miR-296-5p, miR-30b-5p, miR-532-3p, miR-92a-3p, and miR-451a.

The validation cohort samples consisted of 15 AD patients, 14 prodromal AD (very early AD) patients, 22 non-AD MCI subjects, and 21 CTRLs. With these experiments, we confirmed the upregulation of miR-106a-5p, miR-16-5p, miR-223-3p, miR-25-3p, miR-30b-5p, miR-92a-3p, and miR-451a in total EVs from AD patients compared with CTRLs (Figure 3). Moreover, the expression of miR-16-5p, miR-25-3p, miR-92a-3p, and miR-451a was also found to be increased in prodromal AD patients (Figure 3).

A Principal Component Analysis (PCA) for the discovery and validation cohorts was also performed to determine how the miRNA expression data were influenced by the disease. In particular, these components were evaluated using the expression values of the miRNAs that were confirmed to be modulated between the AD and control samples. The *p*-value between the AD and control samples was computed using a Wilcoxon test applied to the first principal component (see Figure 4a,b). Importantly, within the validation cohort (Figure 4b), we were also able to differentiate the control group from all non-control samples along the second principal component (*p* = 4.34 × 10^−5^).

### 2.3. Correlations with Demographics and CSF Biomarkers

The age at sampling was not found to be correlated with any of the miRNAs found to be deregulated in both prodromal AD and AD patients. The gender distribution was not different in any considered group compared with controls, and there were no differences in miRNA levels according to gender. We performed correlation analyses among miRNA levels and CSF biomarkers. Specifically, Spearman correlations between miR-validated miRNAs and clinical features relevant to AD diagnosis (Aβ42, total h-tau, and P-tau181) in prodromal AD and AD patients were analyzed.

Interestingly, we observed, only in the group of AD patients, positive correlations between Aβ42 and miR-30b-5p (r = 0.67, *p* = 0.011) and between h-tau and miR-223-3p (r = 0.62, *p* = 0.026) (Figure 5).

### 2.4. miRNA Target and Pathway Prediction

miRNAs found to be differentially deregulated in prodromal AD and AD patients were further analyzed for their possible target genes and biological pathways using the miRNet platform (https://www.mirnet.ca/miRNet/home.xhtml, accessed on 20 July 2023). The miRNA–target interaction data were downloaded from the miRTarBase v.8 database and were supported by the experimental evidence (Appendix A).

According to the analysis, there were 2810 gene targets and 155 long noncoding RNA (lncRNA) targets for all miRNAs deregulated in prodromal AD and AD patients. Overlaps between gene targets were evident between the combinations of different miRNAs. Therefore, we focused our attention on targets shared among a specific set of analyzed miRNAs. A combination of four miRNAs (miR-16-5p, miR-92a-3p, miR-25-3p, and miR-451a) shared the same two possible targets, named MYC (MYC Proto-Oncogene, BHLH Transcription Factor) and SZRD1 (SUZ RNA Binding Domain Containing 1) (Figure 6), whereas miR-16-5p, miR-92a-3p, and miR-451a shared only three possible targets MYC, SZRD1, and UBE2H (Ubiquitin Conjugating Enzyme E2 H) (Figure 6 and Figure 7b).

Conversely, the combination of miR-16-5p, miR-92a-3p, and miR-25-3p interact with 78 targets, of which 7 are lncRNAs. Among these, XIST (X-inactive Specific Transcript), KCNQ10T1 (KCNQ1 opposite strand/antisense transcript 1), and MEG8 (Maternally expressed 8) are of particular interest. The combination of miR-451a and miR-25 share only two possible gene targets (MYC and SZRD1) and two lncRNA targets NORAD (Non-Coding RNA Activated By DNA Damage) and SNHG17 (Small Nucleolar RNA Host Gene 17).

Among mRNA genes commonly regulated by all deregulated miRNAs (miR-16-5p, miR-92a-3p, miR-25-3p, and miR-451a), we found KLHL15 (Kelch-like protein 15), NOTCH2 (Notch Receptor 2), BTG2 (NGF-inducible anti-proliferative protein PC3), HSPA8 and HSPA1B (heat shock protein family A (Hsp70) member 8 and member 1B), BCL11B (BCL11 Transcription Factor B), NUFIP2 (Nuclear FMR1-interacting protein 2), TRAP1 (TNF Receptor Associated Protein 1), KCNC4 (Potassium Voltage-Gated Channel Subfamily C Member 4), ATOX1 (Antioxidant 1 Copper Chaperone), SMAD7 (SMAD Family Member 7), and UGDH (UDP-Glucose 6-Dehydrogenase), to list a few. Moreover, the Wnt signaling pathway, gap junctions, and glucose metabolism as well as the synaptic vesicle cycle, phagosomes, and focal adhesion were the pathways came to light in this enrichment analysis (Figure 7).

## 3. Discussion

Herein, we showed a deregulated miRNA profile in circulating plasma EVs from AD patients. Specifically, we identified high levels of expression of the following miRNAs in AD patients compared to CTRLs: miR-106a-5p, miR-16-5p, miR-223-3p, miR-25-3p, miR-30b-5p, miR-92a-3p, and miR-451a (Figure 3). Interestingly, the expression levels of the miR-16-5p, miR-25-3p, miR-92a-3p, and miR-451a signature were also found to be increased in prodromal AD, i.e., MCI due to AD, but not in non-AD MCI. Thus, they represent promising candidate biomarkers for the early diagnosis of AD in subjects with mild cognitive decline.

So far, only a few studies have explored the miRNA cargoes in circulating EVs in the context of dementia. The profiles of EV-miRNA expression levels were recently identified from distinct body fluids, including serum, plasma, and CSF, revealing that most of the miRNAs changed in AD target genes related to APP processing, Tau phosphorylation, oxidative phosphorylation, mitochondrial dysfunction, and apoptosis [19,27,28,29,30,31,32]. Interestingly, some of miRNAs found to be differentially expressed in AD patients in this study were already found to be deregulated by other groups [16,33,34,35]. For instance, Cheng et al. (2015) conducted a study where they utilized Next-Generation Sequencing (NGS) and quantitative reverse transcription PCR (qRT-PCR) to validate the differential exosomal miRNA biomarkers between healthy individuals and AD patients. The study identified 15 upregulated miRNAs, including hsa-miR-20a-5p and hsa-miR-106a-5p, that are involved in the pathogenesis of AD. Another study provided the first investigation into the small RNA content of brain-derived EVs and their potential for the early detection of Alzheimer’s disease pathology. In particular, this study identified several deregulated miRNAs, including miR-532-5p, miR-20a-5p, miR-223-3p, miR-17-5p, and miR-19b, which are consistent with our findings [32,34]. Furthermore, Serpente and colleagues, when examining the complete EV fraction, discovered that the patients exhibited modified relative expression levels of numerous miRNAs, including miR-146b-5p, miR-181a-3p, miR-24-3p, miR-125a-5p, let-7b-5p, miR-27a-5p, miR-185-3p, miR-16-5p, miR-15b-5p, miR-30a-5p, and miR-204-5p, in comparison to the CTRLs. Our analysis also revealed a similar deregulation of certain miRNAs [19]. Conversely, some incongruences from these and other studies were probably due to the isolation methods and miRNA sequencing techniques used [16,17,35]. Indeed, Sproviero et al. (2021) provided an objective description of the miRNA profiles found in small and large EVs derived from the plasma of patients with neurodegenerative diseases, indicating unique miRNA signatures that have potential as biomarkers for diagnosis and treatment. However, the authors acknowledged that the limited number of identified deregulated miRNAs in the AD group prevented their pathway classification. Indeed, the researchers detected 33 miRNAs in small EVs and 13 miRNAs in large EVs among AD patients, with 6 miRNAs being distributed in both [17]. In addition, Nie Chao and coworkers (2020) identified eight miRNAs that showed differential expression between AD and control. Of these, three miRNAs (miR-423-5p, miR369-5p, and miR-23a-3p) were significantly upregulated in AD samples compared to control samples, while five miRNAs (miR-204-5p, miR125a-5p, miR-1468-5p, miR-375, and let-7e) were significantly downregulated in AD samples [36]. Also, Lugli et al. showed twenty miRNAs (miR-23b-3p, miR-24-3p, miR-29b-3p, miR-125b-5p, miR-138- 5p, miR-139-5p, miR-141-3p, miR-150-5p, miR-152-3p, miR-185-5p, miR-338-3p, miR342-3p, miR-342-5p, miR-548at-5p, miR-659-5p, miR-3065-5p, miR-3613-3p, miR-3916, miR-4772-3p, and miR-5001-3p) with significant deregulation in the AD group [35]. In contrast to our findings, a separate study discovered certain miRNAs to be differentially downregulated in the plasma EVs of AD patients when compared to controls of similar ages, specifically miR-451a and miR-92a-3p [37]. In summary, the discrepancies that arose across all these studies are likely due to variations in the techniques used to isolate EVs from plasma and the diverse methods for miRNA expression analysis.

MicroRNAs act by regulating gene expression in response to stimuli and during development, and their expression patterns may be implicated in a wide variety of biological processes and various pathologies, including neurological disease [14,16,38]. Consistent with the fact that each miRNA can potentially target hundreds of different genes and thus may affect many biological functions [39], our target prediction analysis for all miRNAs deregulated in prodromal AD and AD patients revealed a broad list of targets: a total of 2810 gene targets and 155 long noncoding RNA (lncRNA) targets. For this reason, to quickly process the huge amount of date obtained, we looked for the functions or pathways on which they converge, choosing exosomes as preferred tissues for our analysis. Interestingly, the miR-16-5p, miR-25-3p, miR-92a-3p, and miR-451a signature shared the same two targets: MYC and SZRD1 (Figure 6).

It is noteworthy that the miRNAs found to be differentially expressed in AD patients belonged to the best-studied miRNA 17-92a cluster (miR-17, miR-19b, miR20a, and miR92a) and its two paralogues: the miR106a-363 (miR-106a, miR-20b, and miR-92a) and miR106b-25 (miR25) clusters. These clusters contained miRNAs that showed a high degree of sequence similarity [25,40], suggesting that these miRNAs regulated a similar set of genes and thus had overlapping functions. The miR17-92a cluster downregulated the expression of the phosphatase and tensin homolog (PTEN) [41], and overall the miR-17-92a, miR-106a-363, and miR-106b-25 clusters regulated the TGF-β signaling pathway and also targeted apoptosis facilitator BCL2L11 [42,43,44,45]. Moreover, members of these clusters (miR-17, miR-20a, and miR-106a) were shown to directly suppress APP [46]. Other studies revealed a direct association of miR-17, miR-19b, miR-20a, and miR106a with aging and age-related conditions [47,48,49]. Furthermore, our study was also in line with recent findings that revealed the deregulation of some miRNAs in AD patients compared to healthy controls as well as using AD mouse models [3,14,50,51,52,53,54,55,56]. It is notable that Wei and his collaborators found that the miR-233 in AD patients was correlated with cognitive decline [54]. Interestingly, we observed a positive correlation between miR-223-3p and total tau levels in AD patients, confirming its possible involvement in the cognitive decline and neurodegeneration occurring in AD. Regarding the correlation analysis of miR-30b-5p, previous evidence demonstrated that upregulated miR-30b caused synaptic and cognitive dysfunction in AD, which we also observed, although the authors suggested that changes in the levels of miR-30b do not influence Aƥ42 processing in patients, which we also confirmed with the positive correlation found between miR-30b-5p and Aƥ-42 levels in AD patients [55]. Other studies had investigated the role of miR-16 in AD and its mechanism of action, demonstrating that genes encoding for APP and SERT (serotonin transport) were identified as two of its targets [56,57,58]. Then, miR-195 was found to modulate the expression of BACE1 [59], whereas miR-451a decreased the expression of neuronal β-secretase 1 in neurons [50]. In addition, a downregulation of miR-451a in the CSF of AD patients, which was correlated with cognition, was also found [50]. Overall, these data suggested that some plasma EV-associated miRNAs might be useful to better characterize AD pathology.

Regarding lncRNA targets, we found KCNQ10T1 (KCNQ1 opposite strand/antisense transcript 1) interacting with three of miRNAs analyzed (miR-16-5p, miR-25-3p, and miR-92a-3p). KCNQ10T1 is related to many diseases, but its exact biological function had not been fully explored [60]. Notably, in addition to be the principal target of the miRNA 17-92 cluster, PTEN was also one of the main targets of KCNQ10T1, supporting the hypothesis of a common regulatory pathway [61]. Another potentially targeted lncRNA was XIST (X-inactive specific transcript), an extensively studied lncRNA associated with several cancer pathologies due to its ability to mediate post-transcriptional gene silencing. Interestingly, both the XIST and KCNQ10T1 lncRNAs bind to PRC2 (Polycomb repressive complex 2), which plays a vital role during development by establishing and maintaining the repressive states of thousands of developmental genes [62,63]. Moreover, some evidence supports the theory that PRC2 function can decline with age, leading to numerous consequences also associated with neurodegeneration [64,65,66]. It is noteworthy that we also observed genes related to different components essential to the PRC2 complex (RBBP7/JARID/EZH1/EZH2) as possible targets for some of our deregulated miRNAs, further supporting the suitability of PRC2 as one of the downstream elements of these hypothetical pathways.

Furthermore, the KEGG functional enrichment analysis revealed microRNAs could drive molecular mechanisms of specific disease-associated pathways in AD. Gap junctions, the regulation of the TGF-beta pathway and the Jak-STAT signaling/Notch signaling pathways, phagosomes, and the synaptic vesicle cycle as well as the neurotrophin signaling pathway, dopaminergic synapses, and lipid metabolism were some of the pathways that came to light in this enrichment analysis (Figure 7). Some of these pathways were associated with molecular and pathological mechanisms in dementia and may be explained by the impairment of the immune system and the atrophy usually found in AD. This evidence was also supported by recent data demonstrating that AD is associated with the inflammatory state and neurotoxicity [46,67,68,69] (Figure 7).

A strength of this study is the CSF confirmation of the AD pathology, whereas a limitation is the lack of samples with other types of dementia, i.e., non-amyloid-driven, to test whether the results are specific to AD. In conclusion, we identified that miR-16-5p, miR-25-3p, miR-92a-3p, and miR-451a were closely related to AD pathology, suggesting a role for these miRNAs in driving neurodegenerative process. We further validated the increased levels of these selected miRNAs in MCI due to AD, named prodromal AD, thus supporting our hypothesis that miR-16-5p, miR-25-3p, miR-92a-3p, and miR-451a represent a novel miRNA signature for a more clearly distinctive diagnosis of AD in the early stage of pathology compared to CTRL and non-AD MCI.

## 4. Materials and Methods

### 4.1. Population and Sample Collection

In total, 26 AD patients, 14 prodromal AD patients, and 41 non-AD MCI subjects were recruited at the Alzheimer’s Unit of Fondazione Ca’ Granda, IRCCS Ospedale Maggiore Policlinico (Milan, Italy).

All patients underwent a standard battery of examinations, including a medical history; a physical and neurological examination; screening laboratory blood tests; a neurocognitive evaluation; MRI imaging; and a lumbar puncture for the quantification of the cerebrospinal fluid (CSF) biomarkers Aβ-amyloid 42 (Aβ42), total tau (h-tau), and tau phosphorylated at position 181 (P-tau181) [70]. The Clinical Dementia Rating (CDR), the Mini-Mental State Examination (MMSE), the Frontal Assessment Battery (FAB), the Wisconsin Card Sorting Test (WCST), and the Tower of London test were used to assess the degree of cognitive impairment. The presence of significant vascular brain damage was excluded (Hachinski Ischemic Score < 4). The diagnoses of non-AD MCI, prodromal AD, and AD were made according to current research criteria [71]. Specifically, the discovery cohort consisted of 11 AD patients (4 males and 7 females, mean age: 75 ± 1 years) and 19 non-AD MCI patients (9 males and 10 females, mean age: 71 ± 9 years). Conversely, the validation cohort consisted of 15 AD patients (7 males and 8 females, mean age: 71 ± 8), 14 prodromal AD patients (4 males and 10 females, mean age: 71 ± 9), and 22 non-AD MCI patients (12 males and 10 females, mean age: 74 ± 6 years). The overall control group consisted of 41 non-demented volunteers matched for ethnic background and age without memory and psychobehavioral dysfunctions (MMSE ≥ 28) recruited at the Geriatric Unit of Fondazione Ca’ Granda, IRCCS Ospedale Maggiore Policlinico (Milan). In particular, considering the discovery cohort, the control group consisted of 20 controls (8 males and 12 females, mean age: 79 ± 6 years), whereas for the validation cohort the control group consisted of 21 patients (6 males and 7 females, mean age: 80 ± 4). This study was approved by the local ethical committee (study No. 5802, approved on 14 September 2021 by Comitato Etico Milano Area 2). All patients and/or their caregivers provided their written informed consent. The characteristics of the patients and controls are summarized in Table 1 and Table 2.

Whole blood samples, collected at the time of diagnosis, were allowed to sit at room temperature for a minimum of 30 min and a max of 2 h and were centrifuged at 2500× *g* for 15 min at room temperature. The plasma supernatant was collected and dispensed in aliquots of 550 µL into cryo-tubes and stored at −80 °C until use.

### 4.2. CSF Processing and Biomarker Analysis

CSF samples were obtained in polypropylene tubes via lumbar puncture at the L4/L5 or L3/L4 interspaces and centrifuged at 4 °C at 2000× *g*. The CSF samples were removed and dispensed in aliquots of 500 µL into cryo-tubes. Specimens were stored at –80 °C until use. Aβ42, h-tau, and P-tau CSF levels were determined with Lumipulse, as previously reported [70].

### 4.3. Isolation of Total Vesicles and Characterization

Aliquots of 250 µL of plasma were centrifuged at 10,000× *g* for 10 min to remove cells and debris. The supernatant was transferred to a sterile tube and incubated for 5 min at room temperature with 2.5 µL of purified thrombin (500 U/mL, System Bioscience, Palo Alto, CA, USA). This step was required to remove the large amount of fibrin present in the plasma. After incubation for 5 min, samples were centrifuged at 10,000× *g* for 5 min. Thereafter, the supernatants were transferred into new tubes, and 67 µL of Exoquick^®^ exosome precipitation solution (System Biosciences, Inc., Mountainview, CA, USA) was added, gently mixed by inversion, and incubated for 60 min at 4 °C. The resulting suspension was centrifuged at 3000× *g* for 10 min at room temperature to obtain pellets containing total EVs. These pellets were then re-suspended in 400 µL of Buffer B and Buffer A of ExoQuick Ultra at a ratio of 1:1. The entire content was loaded into a pre-cleaned resin column with another 100 μL of Buffer B that had previously been added. After mixing, purified EVs were collected via centrifugation at 1000× *g* for 2 min. All separated total EVs were aliquoted and stored at −80 °C until use.

The size distribution and concentration of the particles in the total EVs were measured with NanoSight (NS300) (Malvern Panalytical Ltd., Malvern, UK) equipped with nanoparticle tracking analysis (NTA) software (NanoSight NS300; Malvern Panalytical Ltd., Malvern, UK).

To lyse total EVs, each tube received 1 volume of M-PER mammalian protein extraction reagent (Thermo Scientific, Inc., Waltham, MA, USA) containing protease and phosphatase inhibitors. Aliquots of EV lysates were separated using SDS-PAGE on 4-15% acrylamide gels, and proteins were transferred to PVDF membranes. After blocking for 2 h with 5% BSA in TBS, the membranes were incubated overnight at 4 °C with the desired primary antibodies diluted in TBS. In the present study, the following antibodies and dilutions were used: anti-TGS101 from abcam at 1:500 (ab125011); and anti-CD81 and anti-GOLGI2/GM130 from Fine Test, both used at 1:500 (032FNFNab01501 (anti-CD81 antibody) and 032FNFNab03558 (anti-GOLGA2/GM130 antibody)). Membranes were washed and incubated with an appropriate peroxidase-conjugated secondary antibody (1:10,000 dilution), and proteins were visualized with a chemiluminescence reaction. All the experiments reported in this study were repeated at least three times, and comparable results were obtained. The blots reported in the figures are representative images. Images of reactive bands were acquired using an Odyssey Fc imager (LI COR).

Specimens for transmission electron microscopy (TEM) were prepared using the conventional negative stain protocol. Briefly, a small drop of a sample was adsorbed onto a formvar-coated copper grid, washed with two drops of deionized water, and stained with a drop of 2% uranyl acetate in bidistilled water. Samples were observed using a Talos L120C transmission electron microscope (Thermo Fisher Scientific, Waltham, MA, USA) equipped with a 4 K Ceta CMOS digital camera (Thermo Fisher Scientific). Twenty microscopic fields were observed, and representative images were selected.

### 4.4. RNA Extraction and Quality Control

Total RNA, including small RNAs, was purified from total EVs by exploiting an miRNeasy Micro Kit (Qiagen, Hilden, Germany) that used QIAzol Lysis Reagent/Chloroform extraction to provide a robust RNA purification step. Then, 1.5 volumes of 100% ethanol was added to samples that were passed through an RNeasy MinElute spin column that immobilized the RNA. The RNeasy MinElute spin column was washed, and the RNA was eluted with 15 μL of RNase-free water according to the manufacturer’s instructions. The RNA from total vesicles was analyzed using an Agilent 2100 Bioanalyzer and an RNA 6000 Pico kit (Agilent Technologies, Santa Clara, CA, USA) for the determination of the RNA concentration, purity, and integrity. The RNA amount was represented by a peak spanning 25–200 nts.

### 4.5. Expression Analysis of microRNA Using TaqMan^®^ OpenArray^®^ MicroRNA Panels

The RNA from total EVs was reverse-transcribed using a TaqMan MicroRNA Reverse Transcription Kit (Applied Biosystem, Waltham, MA, USA) with Megaplex RT Primers PoolA and PoolB. Given the low RNA amounts obtained in the extraction step, cDNA was preamplified prior to the final real-time PCR step. The preamplification step was performed using TaqMan PreAmp Master Mix and the Megaplex PreAmp Primers PoolA and PoolB. The preamplified cDNA (diluted 1:20) was mixed with TaqMan OpenArray Real-Time Master Mix to perform real-time PCR on an OpenArray plate. A 5 μL sample of the PCR reaction was added to each well of a 384-well plate. Then, samples with the master mix were loaded from the 384-well sample plate onto the OpenArray plate using the OpenArray AccuFill System. Finally, PCR was run on the QuantStudio 12K Flex Real-Time PCR System (Thermo Fisher Scientific, Waltham, MA, USA).

### 4.6. Validation Analysis with Taqman MicroRNA Assays

MiRNAs found to be deregulated were validated with TaqMan^®^ microRNA Assays (Applied Biosystem). For miRNAs belonging to the same miRNA cluster, we selected only a few candidate miRNAs, which are listed below: miR-106a-5p, miR-16-5p, miR-223-3p, miR-25-3p, miR-296-5p, miR-30b-5p, miR-532-3p, miR-92a-3p, and miR-451a. Briefly, 3 μL of total RNA was specifically retrotranscribed using custom RT primer pools. Again, the cDNA was first preamplified, and then PCR was run on a QuantStudio 12k Flex system using the following PCR protocol: one cycle of enzyme activation at 95 °C for 10 min, 45 cycles of denaturation at 95 °C for 15 s, and annealing/extension at 60 °C for 60 s.

### 4.7. Target Prediction and Pathway Enrichment Analysis

The MiRNet (https://www.mirnet.ca/miRNet/home.xhtml, accessed on 20 July 2023) web tool was used to provide visual exploration and functional interpretation of the miRNA–target interaction network and a pathway enrichment analysis [34]. A functional enrichment analysis was performed using the KEGG database with two different algorithms implemented in the miRNet tool (hypergeometric tests and empirical sampling, as recently proposed [35]).

miRNet 2.0 supports four query types, two enrichment algorithms (hypergeometric tests and empirical sampling), and nine annotation libraries for functional enrichment analysis that include the following: gene ontology (GO), Kyoto Encyclopedia of Genes and Genomes (KEGG), Reactome, and disease ontology databases. The miRNA set libraries are based on the TAM 2.0 database, which includes miRNA-function, miRNA-disease, miRNA-TF, miRNA-cluster, miRNA-family, and miRNA-tissue set libraries [38].

### 4.8. Statistical Analysis

Normalized Ct values of miRNAs were used to analyze differences between healthy subjects and patients. In the discovery phase, the statistical significance of miRNA modulation was assessed using the Wilcoxon rank sum test. The analysis was carried out with Matlab R2022a. Regarding the validation phase, the statistical analysis was performed using GraphPad Prism 9.0 Software (GraphPad Software Inc., San Diego, CA, USA). The data distribution was tested for normality with the Kolmogorov–Smirnov and Shapiro–Wilk tests. Multiple comparisons were performed using a one-way ANOVA followed by Tukey’s test or the Kruskal–Wallis test followed by Dunn’s Multiple Comparison test as post hoc tests. The Spearman test was applied for correlations between deregulated miRNAs and age or clinical variables and pathogenic CSF protein levels. Significance was defined at the 5% level, and all data are shown as means ± SEMs. Lastly, a chi-squared test was used for the gender distribution among the groups.

## Figures and Tables

**Figure 1 ijms-24-14749-f001:**
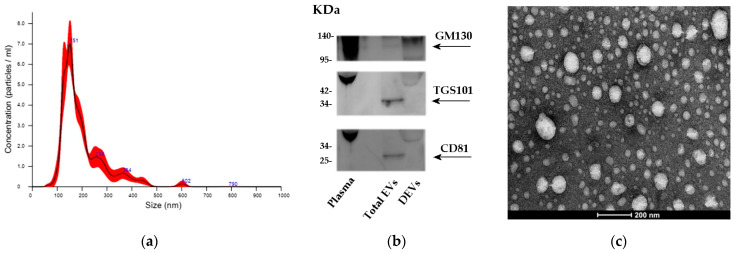
Characterization of isolated plasma vesicles. (**a**) Representative trace of nanoparticle tracking analysis (NTA) of total EVs isolated using ExoQuick ULTRA. (**b**) Plasma, total EVs, and depleted EVs (DEVs) were analyzed by immunoblotting with antibodies against CD81, TGS101, and GM130, as indicated on the right. (**c**) TEM images revealed the peculiar oval shape of circulating EVs.

**Figure 2 ijms-24-14749-f002:**
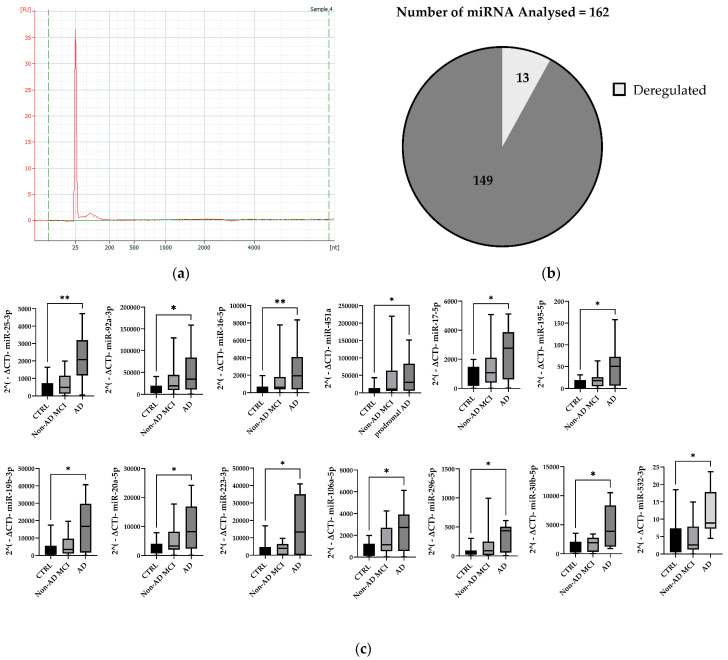
MiRNA expression profile in total plasma EVs. (**a**) Concentration of EV-derived miRNA after using miRNeasy Isolation Kit assessed by Bioanalyzer. (**b**) Pie chart showing deregulated miRNAs from the total number of miRNAs analyzed. (**c**) Box plots showing 2^−ΔCt^. and min and max values for CTRL, non-AD MCI, and AD patients. * *p* < 0.05; ** *p* < 0.005.

**Figure 3 ijms-24-14749-f003:**
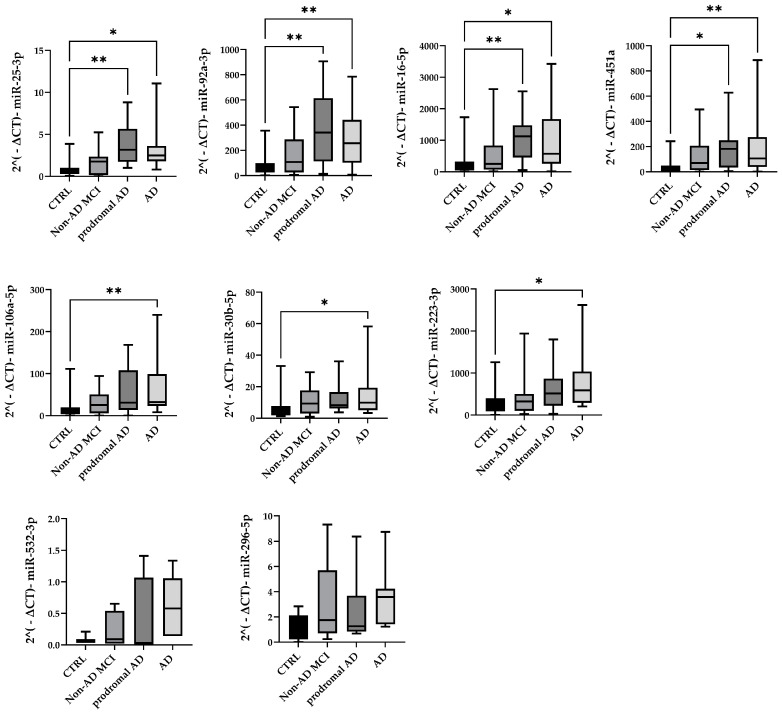
miRNA expression levels in non-AD MCI, prodromal AD, AD, and CTRL, shown as 2^−ΔCt^. * *p* < 0.05; ** *p* < 0.005.

**Figure 4 ijms-24-14749-f004:**
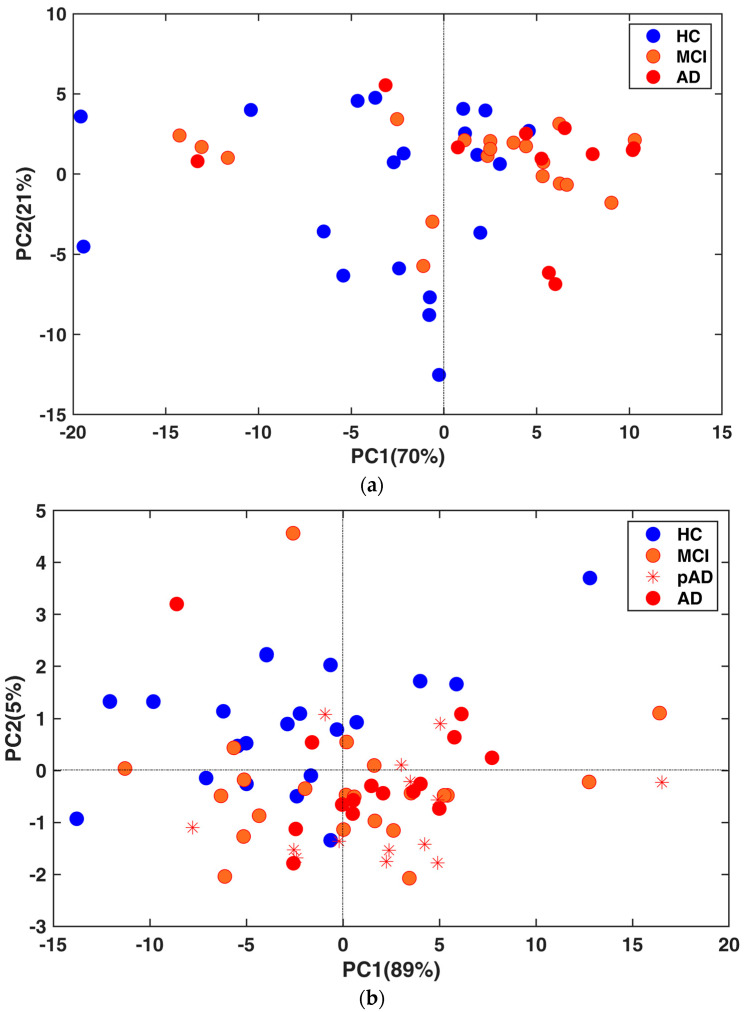
(**a**) PCA based on the most variant miRNAs in the discovery cohort (*p* = 0.004). (**b**) PCA based on the most variant miRNAs in the validation cohort (*p* = 0.007). The *p*-value between AD and control samples was computed using a Wilcoxon test applied to the first principal component.

**Figure 5 ijms-24-14749-f005:**
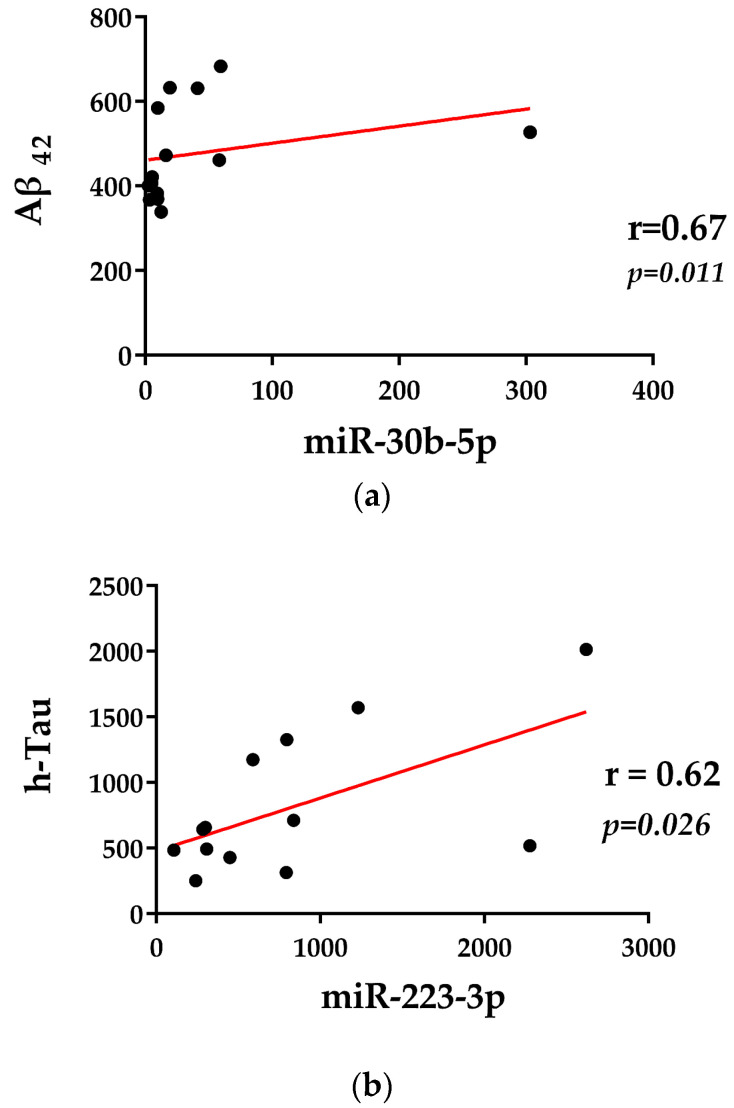
(**a**) The Spearman correlation analysis between miR-30b-5p and Aβ_42_. (**b**) The Spearman correlation analysis between miR-30b-5p and Aβ_42_.

**Figure 6 ijms-24-14749-f006:**
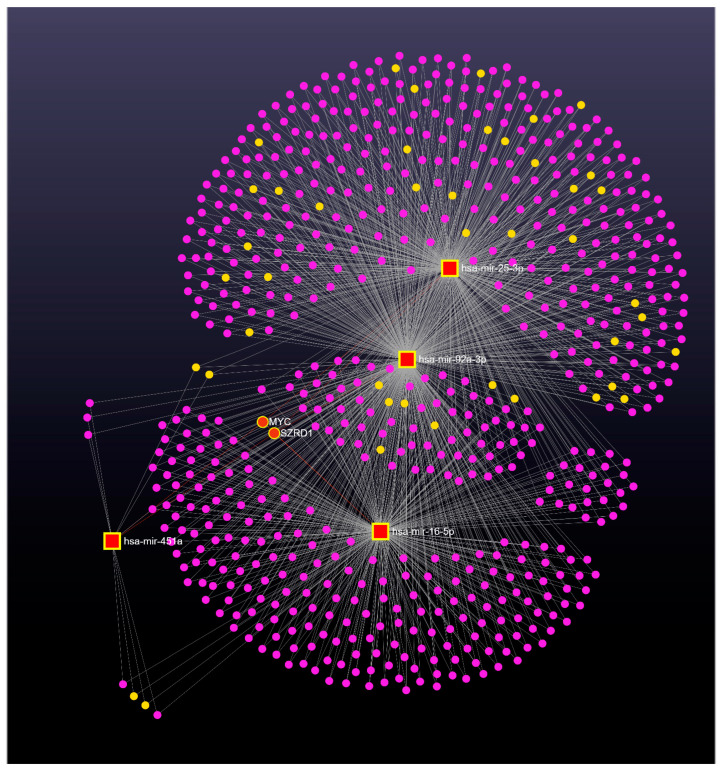
Network visualization for the enrichment analysis of miR-92a-3p, miR-16-5p, miR-25-3p, and miR-451a in prodromal AD and AD groups. Red/yellow squares represent deregulated miRNAs and pink circles are their target genes, while yellow circles are their lncRNA targets. Big red/yellow circles represent shared interactions.

**Figure 7 ijms-24-14749-f007:**
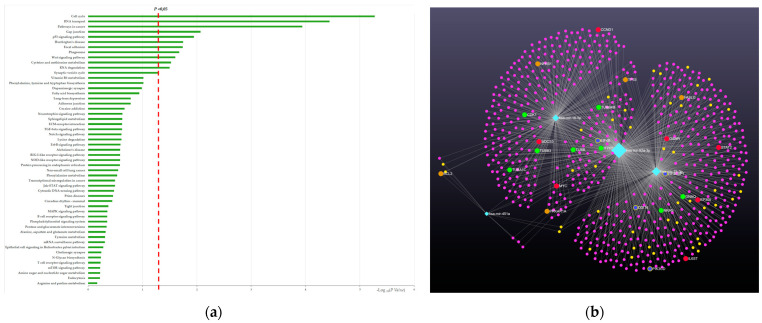
MiRNA target and pathway prediction. (**a**) Biological pathway (y-axis) associated with mir-25-3p, mir-92a-3p, mir-16-5p and mir-451a signature. The significance of this association is expressed with –log10 (*p*-value) and is reported on x-axis. (**b**) Network visualization for the enrichment analysis of miR-92a-3p, miR-16-5p, miR-25-3p, and miR-451a in prodromal AD and AD group. Pink circles are their targets genes while yellow circles are their lncRNA targets. Big orange circles represent Wnt Signaling pathway; big blue circles show genes involved in phagosomes; red circles show gap-junction-associated genes; green circles represent genes related to synaptic vesicle cycle.

**Table 1 ijms-24-14749-t001:** Discovery Cohort.

Variable	AD	Non-AD MCI	CTRL
N	11	19	20
Gender (M:F)	4:7	9:10	8:12
Mean aβ42 ± SEM (pg/mL)	551 ± 1.79	869.63 ± 2.48	-
Mean h-tau ± SEM (pg/mL)	842.64 ± 58	434.74 ± 3.7	-
Mean p-tau ± SEM (pg/mL)	95.18 ± 1.18	72.53 ± 1.5	-
Mean Age (years ± SD)	75 ± 1	75 ± 6	79 ± 6
MMSE (mean ± SD)	18.30 ± 5.58	26.37 ± 2.61	28.15 ± 1.35

**Table 2 ijms-24-14749-t002:** Validation Cohort.

Variable	AD	Prodromal AD	Non-AD MCI	CTRL
N	15	14	22	21
Gender (M:F)	7:8	4:10	12:10	6:15
Mean aβ42 ± SEM (pg/mL)	479.47 ± 1.31	518.79 ± 1.67	916.14 ± 2.33	-
Mean h-tau ± SEM (pg/mL)	829.40 ± 4.89	661.36 ± 4.17	410.24 ± 3.00	-
Mean p-tau ± SEM (pg/mL)	94.13 ± 1.01	112.57 ± 1.55	64.38 ± 1.30	-
Mean Age (years ± SD)	71 ± 8	71 ± 9	74 ± 6	80 ± 4
MMSE (mean ± SD)	18.36 ± 6.98	26.77 ± 1.96	26.68 ± 2.12	28.15 ± 1.27

**Table 3 ijms-24-14749-t003:** Log of fold change (FC) values of deregulated miRNA and *p*-values between AD and CTRL.

miRNA	Log FC	*p*-Value
hsa-miR-16-5p	3.058795	0.006041
hsa-miR-92a-3p	2.339818	0.007746
hsa-miR-106a-5p	2.251733	0.006168
hsa-miR-451a	2.875201	0.012556
hsa-miR-19b-3p	2.642532	0.005323
hsa-miR-17-5p	2.078854	0.009754
hsa-miR-223-3p	4.468833	0.015466
hsa-miR-296-5p	1.851545	0.014716
hsa-miR-20a-5p	2.042569	0.014164
hsa-miR-195-5p	2.541182	0.016437
hsa-miR-532-3p	2.709	0.018207

## Data Availability

The data that support the findings of this study are available from the corresponding author upon reasonable request.

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
