# Peer review of "Altered Extracellular Vesicle miRNA Profile in Prodromal Alzheimer’s Disease"

_ijms, 2023, doi:10.3390/ijms241914749_

Round 1
Reviewer 1 Report
we read with interest the article by Visconte et al aimed to assess miRNA from subcohort with AD, prodromal AD, non-AD MCI, and CTRLs; however, when the data were presented showed that the only deregulated miRNAs from the AD cohort were focused on, and were analyzed.
Comments:
The study should incorporate the deregulated miRNA in the different cohorts to establish a meaningful differential miRNA profile and altered miRNA pathways.
the authors should also perform the profiled miRNA cluster in PCA analysis to show the variations among samples and how distinct the different testing groups are from the controls..
3- Please include the detailed methodology of how the authors build the miRNA pathways with miRNA, and what parameters and variables were used in the analysis.
4-Please indicate what software was used to perform the GO enrichment from the KEGG database as there are several open sources; however, they provide different results, thus it is important to indicate which one the authors used.
5- Can the author explain how they reached the 162 miRNA if they started with 754 human miRNAs with having 12% of all detectable miRNAs as significant and how 13 are deregulated?
6- The authors should provide the raw data of each sample readout so as to allow other researchers to perform secondary analysis.
Author Response
“The study should incorporate the deregulated miRNA in the different cohorts to establish a meaningful differential miRNA profile and altered miRNA pathways. “
We thank the reviewer for this comment, we agree that the final aim of the study should be to incorporate the deregulated miRNA in the different cohorts. Nevertheless, we would like to be cautious in this very first stage of the research as 1) prodromal AD may have different pathogenic mechanisms as compared with full blown AD 2) non-AD MCI cannot be considered controls as they can develop other non-AD dementia over time. Therefore, we preferred not to merge groups in this study, but the longitudinal follow up is ongoing in order to rule out, in the future, whether 1) the profile of prodromal AD will change when the will develop full-blown dementia 2) non-AD MCI will revert to a normal status and can be thus merged with controls.
“The authors should also perform the profiled miRNA cluster in PCA analysis to show the variations among samples and how distinct the different testing groups are from the controls.”
We thank the reviewer for this suggestion. In response, we have included the Principal Component Analysis (PCA) results for both the discovery and validation cohorts in the new Fig. 4A and Fig 4B. These components were evaluated using the expression values of the miRNAs that were confirmed to be modulated between AD and Control samples. The p-value between AD and control samples was computed using a Wilcoxon test applied to the first principal component (see Fig 4A and Fig4B). Importantly, within the validation cohort (Fig4B), we were also able to differentiate the control group from all non-control samples along the second principal component (p = 4.34e-05). This part in now incorporated in the results section as follows:
“Principal Component Analysis (PCA) for the discovery and validation cohorts was also performed, to determinate how miRNA expression data were influenced by the disease. In particular, these components were evaluated using the expression values of the miRNAs that were confirmed to be modulated between AD and Control samples. The p-value be-tween AD and control samples was computed using a Wilcoxon test applied to the first principal component (see Fig4a and Figab). Importantly, within the validation cohort (FigXX2), we were also able to differentiate the control group from all non-control samples along the second principal component (p = 4.34e-05).”
Page 6, lines 1-8, page 7
“3-Please include the detailed methodology of how the authors build the miRNA pathways with miRNA, and what parameters and variables were used in the analysis.”
As the reviewer suggested we improved our explanation about the use of miRNet tool in the methods section and rephrased the paragraph as follows:
“The MiRNet (https://www.mirnet.ca/miRNet/home.xhtml, accessed on 20 July 2023) web tool was used to provide visual exploration and functional interpretation of miRNA-target interaction network and a pathways enrichment analysis [34]. Functional enrichment analysis was performed using the KEGG database with two different algorithms implemented in the miRNet tool: hypergeometric tests and empirical sampling, as recently proposed [35]” Methods section page 16, lines 4-9
“4-Please indicate what software was used to perform the GO enrichment from the KEGG database as there are several open sources; however, they provide different results, thus it is important to indicate which one the authors used.”
We thank the reviewer for this suggestion, we tried to better explain how the software miRnet2.0 performs the functional enrichment analysis as follows:
“miRNet 2.0 supports four query type, two enrichment algorithms (hypergeometric yests and empirical sampling), and nine annotation libraries for functional enrichment analysis that inlude the following: gene ontology (GO), Kyoto Encyclopedia of Genes and Genomes (KEGG), Reactome, and disease ontology databases. The miRNA set libraries are based on TAM 2.0 database, which includes miRNA-function, miRNA-disease, miRNA-TF, miRNA-cluster, miRNA-family, and miRNA-tissue set libraries [38]. Methods section page 16, lines 10-15
“5- Can the author explain how they reached the 162 miRNA if they started with 754 human miRNAs with having 12% of all detectable miRNAs as significant and how 13 are deregulated?”
We apologized to the reviewer for the lack of clarity. It is well-established that the detection rate for miRNAs in samples from liquid biopsy is significantly lower than that obtained from tissue samples. To ensure sufficient statistical power, we took the step of excluding miRNAs that were undetectable in more than 5 samples for each subgroup from further analysis. After this filtering process, 162 miRNAs that were expressed in at least 80% of the total samples remained. This approach allowed us to focus our statistical analyses on the most reliable miRNAs. To assess significant differences in the modulation of the expressed miRNAs, we then applied non-parametric statistical tests for multiple group comparisons. Among the 13 miRNAs depicted in Figure 2C, we observed significant modulation between AD and control samples. In Figure 3, we show the miRNAs that were validated as being modulated out of the 13 in a separate cohort.
We added few lines as follows in the results section: Specifically, to ensure sufficient statistical power, we took the step of excluding miRNAs that were undetectable in more than 5 samples for each subgroup from further analysis. After this filtering process, 162 miRNA expressed in at least 80% of the total samples remained. This approach allowed us to focus our statistical analyses on the most reliable miRNAs. To assess significant differences in the modulation of the expressed miRNAs, we then applied non-parametric statistical tests for multiple group comparisons. Out of the 162 miRNAs analyzed, 13 were found to be deregulated in AD patients compared to CTRL. Results section page 4 lines 5-12.
“6- The authors should provide the raw data of each sample readout so as to allow other researchers to perform secondary analysis.” We agree with the reviewer and implemented the manuscript with the raws data of the discovery cohort analysis now provided in suppl Figure 1.
Reviewer 2 Report
The authors study the miRNA profile in exosome from Alzheimer’s disease patients, which can be potential biomarkers and targets for AD. This is an interesting study with future clinical applications.
minor issues:
1, the authors have show the table 1 and table 2 with the title as "discovery cohort" and "validation cohort". However, authors have not described the detailed information on these. they should provide these information.
2, the authors have also the data of AB 42, P-TAU levels of patients. So can I suggest to do some correlation analysis to see whether miRNA profile can be linked with AB42 and p-tau levels in patients.
3, the authors should also need to compare and discuss their findings with previous reports on exosome miRNA profile in AD in the discussion section.
Author Response
“1, the authors have show the table 1 and table 2 with the title as "discovery cohort" and "validation cohort". However, authors have not described the detailed information on these. they should provide these information. “
We thank the reviewer for this obervation and according to the suggestion we implemented the text with the description of the discovery and validation cohorts in the methods section as follows:
Specifically, the discovery cohort consisted of 11 AD patients (4 males:7 females, mean age 75 yrs ±1), 19 non-AD MCI patients ( 9 males and 10 females, mean age 71 yrs ± 9), Conversely, the validation cohort consisted of 15 AD patients ( 7 males and 8 females, mean age: 71±8), 14 Prodromal AD ( 4 males and 10 females, mean age: 71±9), 22 non-AD MCI ( 12 males and 10 females, mean age: 74 yrs±6) The overall control group consisted of 41 non-demented volunteers matched for ethnic background and age and without memory and psycobehavioural dysfunctions (MMSE ≥ 28) recruited at the Geriatric Unit of Fondazione Ca’ Granda, IRCCS Ospedale Maggiore Policlinico (Milan). In particular considering the discovery cohort, the control group consisted of 20 controls (8 males and 12 females, mean age: 79 yrs ±6 ), whereas for the validation cohort the controls group consisted of 21 patients ( 6 males: 7 females, mean age: 80±4). Methods Page 14, lines 10-20.
“2, the authors have also the data of AB 42, P-TAU levels of patients. So can I suggest to do some correlation analysis to see whether miRNA profile can be linked with AB42 and p-tau levels in patients.”
We thank the reviewer for this comment and as suggested we extended our correlation analysis to all the miRNAs validated , we rephrased, implemented this part as follows and added figures 5a and 5b:
We performed correlation analyses among miRNAs levels and CSF biomarkers. Specifically Spearman correlations between miR-validated miRNAs and clinical features relevant to AD diagnosis (Aβ42, total h-tau and P-tau181) in prodromal AD and AD patients were analyzed. Interestingly, we observed only in the group of AD patients a positive correlation between Aβ42 and miR-30b-5p (r=0.67, p=0.011) and between h-tau and miR-223-3p (r=0.62, p=0.026) ( Figure 5) Page 8, lines 4-10.
“3, the authors should also need to compare and discuss their findings with previous reports on exosome miRNA profile in AD in the discussion section.”
As suggested by the reviewer, we have expanded the result findings in the discussion section in the light of recent literature regarding exome miRNA profile in AD. This new part has been added in the discussion as follows:
For instance, Cheng et al. (2015) conducted a study where they utilised Next Generation Sequencing (NGS) and quantitative reverse transcription PCR (qRT-PCR) to validate the differential exosomal miRNA biomarkers between healthy individuals and AD patients. The study identified 15 upregulated miRNAs, including hsa-miR-20a-5p and hsa-miR-106a-5p, that are involved in the pathogenesis of AD. Another study provided the first investigation into the small RNA content of brain-derived EVs and their potential for early detection of Alzheimer's disease pathology. In particular, this study identified several deregulated miRNAs, including miR-532-5p, miR-20a-5p, miR-223-3p, miR-17-5p, and miR-19b, which are consistent with our findings [32,34]. Furthermore, Serpente and colleagues, when examining the complete EVs fraction, discovered that the patients exhibited modified relative expression levels of numerous miRNAs including miR-146b-5p, miR-181a-3p, miR-24-3p, miR-125a-5p, let-7b-5p, miR-27a-5p, miR-185-3p, miR-16-5p, miR-15b-5p, miR-30a-5p, and miR-204-5p in comparison to the CTRLs. Our analysis also revealed similar deregulation of certain miRNAs [19]. Conversely, some incongruences from these and other studies were probablydue to the isolation methods and miRNA sequencing techniques used [16,17,35]. Indeed, Sproviero et al. (2021) provided an objective description of miRNA profiles found in small and large EVs derived from the plasma of patients with neurodegenerative diseases, indicating unique miRNA signatures that have potential as biomarkers for diagnosis and treatment. However, the authors acknowledged that the limited number of identified deregulated miRNAs in the AD group prevented their pathway classification, indeed the researchers detected 33 miRNAs in Small-EVs and 13 in Large-EVs among AD patients, with six miRNAs being distributed in both [17]. In addition, Nie Chao and coworkers (2020) identified eight miRNAs that showed differential expression between AD and Control. Of these, three miRNAs (miR-423-5p, miR369-5p, and miR-23a-3p) were significantly upregulated in AD samples compared to Control samples, while five miRNAs (miR-204-5p, miR125a-5p, miR-1468-5p, miR-375, and let-7e) were significantly downregulated in AD samples [36]. Also Lugli et al., showed twenty miRNAs (miR-23b-3p, miR-24-3p, miR-29b-3p, miR-125b-5p, miR-138- 5p, miR-139-5p, miR-141-3p, miR-150-5p, miR-152-3p, miR-185-5p, miR-338-3p, miR342-3p, miR-342-5p, miR-548at-5p, miR-659-5p, miR-3065-5p, miR-3613-3p, miR-3916, miR-4772-3p, miR-5001-3p) with significant deregulation in AD group [35]. In contrast to our findings, a separate study discovered certain miRNAs to be differentially downregulated in the plasma EVs of AD patients when compared to controls of similar age, specifically miR-451a and miR-92a-3p [37]. In summary, the discrepancies that arose across all these studies are likely due to variations in the techniques used to isolate EVs from plasma and the diverse methods for miRNAs expression analysis. Discussion section, page 11, lines 14-50
Round 2
Reviewer 1 Report
accept